# Increase in Tuberculosis Diagnostic Delay during First Wave of the COVID-19 Pandemic: Data from an Italian Infectious Disease Referral Hospital

**DOI:** 10.3390/antibiotics10030272

**Published:** 2021-03-08

**Authors:** Francesco Di Gennaro, Gina Gualano, Laura Timelli, Pietro Vittozzi, Virginia Di Bari, Raffaella Libertone, Carlotta Cerva, Luigi Pinnarelli, Carla Nisii, Stefania Ianniello, Silvia Mosti, Nazario Bevilacqua, Fabio Iacomi, Annalisa Mondi, Simone Topino, Delia Goletti, Francesco Vaia, Giuseppe Ippolito, Enrico Girardi, Fabrizio Palmieri

**Affiliations:** 1National Institute for Infectious Diseases “L. Spallanzani” IRCCS, 00161 Rome, Italy; gina.gualano@inmi.it (G.G.); laura.timelli@inmi.it (L.T.); pietro.vittozzi@inmi.it (P.V.); virginia.dibari@inmi.it (V.D.B.); Raffaella.libertone@inmi.it (R.L.); Carlotta.cerva@inmi.it (C.C.); carla.nisii@inmi.it (C.N.); stefania.ianniello@inmi.it (S.I.); silvia.mosti@inmi.it (S.M.); nazario.bevilacqua@inmi.it (N.B.); Fabio.iacomi@inmi.it (F.I.); annalisa.mondi@inmi.it (A.M.); simone.topino@inmi.it (S.T.); delia.goletti@inmi.it (D.G.); Francesco.vaia@inmi.it (F.V.); giuseppe.ippolito@inmi.it (G.I.); enrico.girardi@inmi.it (E.G.); fabrizio.palmieri@inmi.it (F.P.); 2Department of Epidemiology of Lazio Regional Health Service, 00147 Rome, Italy; l.pinnarelli@deplazio.it

**Keywords:** tuberculosis, diagnostic delay, COVID 19, SARS CoV2, pandemic

## Abstract

Background: The WHO advised that the impact of COVID-19 pandemic on TB services was estimated to be dramatic due to the disruption of TB services. Methods: A retrospective data collection and evaluation was conducted to include all the patients hospitalized for TB at INMI from 9 March to 31 August 2020 (lockdown period and three months thereafter). For the purpose of the study, data from patients hospitalized in the same period of 2019 were also collected. Results: In the period of March–August 2019, 201 patients were hospitalized with a diagnosis of TB, while in the same period of 2020, only 115 patients, with a case reduction of 43%. Patients with weight loss, acute respiratory failure, concurrent extrapulmonary TB, and higher Timika radiographic scores were significantly more frequently hospitalized during 2020 vs. 2019. The median patient delay was 75 days (IQR: 40–100) in 2020 compared to 30 days (IQR: 10–60) in 2019 (*p* < 0.01). Diagnostic delays in 2020 remain significant in the multiple logistic model (AOR = 6.93, 95%CI: 3.9–12.3). Conclusions: Our experience suggests that COVID-19 pandemic had an impact on TB patient care in terms of higher diagnostic delay, reduction in hospitalization, and a greater severity of clinical presentations.

## 1. Introduction

COVID-19, as a result of severe acute respiratory syndrome coronavirus 2 (SARS-CoV-2) infection, has been the direct cause of hundreds of thousands of deaths in the world [1,2]. The direct and indirect effects of the pandemic, acting through social, economic, environmental and healthcare pathways, can also be countless [2].

During the first wave of the pandemic, most countries from all over the world declared national lockdown as containment and mitigation measure in order to slow down the virus circulation [3]. Prevention and treatment services for noncommunicable diseases (NCDs) have been severely disrupted since the COVID-19 pandemic began, according to a WHO survey [4]

In fact, the burden on health services caused by the COVID-19 emergency has led to several changes in the ordinary management of both communicable and non-communicable diseases, following the reduction or suspension of non-urgent outpatient care (for example, blocking of visits and non-urgent surgery) [5,6].

The impact of COVID-19 pandemic on TB services was estimated to be dramatic, especially in countries where healthcare staff involved in TB management have been reassigned to the COVID-19 emergency [6,7]. WHO advised that the disruption of TB services, due to the COVID-19 pandemic, could lead to fewer TB diagnoses, an increase in diagnostic delay, and TB death [8]. However, few data are yet available on what actually happened [9]. Diagnostic delay with more severe clinical presentations, worst outcomes, and lost to follow-up, are only few of the indirect effects of health services disruption expected mainly during the acute phases of the pandemic [10,11]. Italy was the first European country to be affected by COVID-19 and the first to implement a national lockdown measure along with a strong health services reorganization [12]. In fact, between 9 March and 18 May 18 2020, a national containment and mitigation strategy focused on lockdown was implemented to flatten the curve of the COVID-19 pandemic and to reduce the stress on the Italian National Health Service (INHS) [13].

In the same timeframe, INHS was forced to reorganize itself with the conversion of most hospital wards to COVID-19 units, the health workforce was reallocated, and most of the non-urgent services were suspended [14]. In this scenario, most wards and outpatient services for TB of Italian hospital were also interrupted in their routine activities, which is what happened at the National Institute for Infectious Diseases “L.Spallanzani” (INMI) in Rome, an Italian TB referral hospital.

With the purpose to evaluate the potential impact of first wave of COVID-19 pandemic on TB cases, we performed a retrospective, observational cohort analysis of socio-economic aspects, clinical, microbiological, and radiological findings of patients hospitalized for TB at INMI during 2019 and 2020.

## 2. Results

### 2.1. Clinical, Microbiological and Radiological Findings

In the period of March–August 2019, 201 patients were hospitalized with a diagnosis of TB, while in the same period of 2020 only 115 patients, with a case reduction of 43% Figure 1 shows the number of TB cases by month of hospitalization, both in 2019 and 2020. We observed lower cases during the months of March and April 2020 compared with the same months of 2019; this difference tended to reduce in the months of May, June, and July, and then seemed to increase again during the month of August. The population characteristics of two periods of study are reported in Table 1. During the period included in 2020 more patients with a low BMI were observed, compared to 2019 (57% vs. 34% *p* < 0.01); patients were more unemployed, 63% vs. 46% (*p* = 0.013); less instructed (92% with <8 years of education, vs. 44%: *p* = 0.038); more smokers, 56% vs. 35% (*p* < 0.01). Four TB patients were also coinfected with SARS-CoV-2.

The clinical characteristics are shown in Table 2. During 2020 vs. 2019 period patients with dyspnea (23% vs. 8%), weight loss (46% vs. 28%), with acute respiratory failure (30% vs. 8%) and concurrent extra pulmonary TB (32% vs. 15%), were significantly more frequently hospitalized.

Timika score showed a significant worse radiographic pattern in 2020 patients vs. the 2019 group. (*p* = < 0.01).

### 2.2. Diagnostic Delays

A significant increase in total diagnostic delay (TD, time interval between the onset of symptoms and start of TB treatment) was observed. The median patient delay (PD, period from the onset of the first symptom(s) related to pulmonary TB to the first medical consultation) was 75 days (IQR: 40–100) in 2020 compared to 30 days (IQR: 10–60) in 2019 (*p* < 0.01). The delay by month of admission in 2020 was higher for all the months considered, except for March, compared with the same months of 2019 (Figure 2). The healthcare system (HSD, the time interval between the first medical consultation and start of TB treatment) delay was 5 days (IQR: 3–14) in 2020 versus 4 (IQR: 2–10) in 2019; the total delay (PD+HSD) was 90 days (IQR: 58–107) and 38 (IQR: 22–69), in 2020 and in 2019, respectively (*p* < 0.01) (Table 3).

As Table 4 shown low education (<8 years), foreign nationality, year 2020, BMI < 18.5, and acute respiratory failure, were found to be linked with long PD.

As shown in Table 5, diagnostic delay in 2020 remain significant in the multiple logistic model (AOR = 6.93, 95%CI: 3.91–12.30) also when adjusted for gender, education > 8 years, foreign nationality, occupational status, BMI, the presence of comorbidity, and TB risk factors

## 3. Discussion

In this retrospective study, we evaluated socio-economic aspects, clinical, microbiological, and radiological findings of TB patients admitted during two different periods (2019 vs. 2020) to INMI. Our data showed a reduction of hospitalization for TB, with a significative increase of both total and patients delay during 2020 period vs. 2019. In addition, a trend towards greater severity of clinical presentations was observed in 2020. Evaluating hospitalization for TB in our hospital over the period March–August 2020 with the previous year, a significative reduction in hospitalizations—up to 40%—emerged. This evidence is consistent with data form other Italian [15,16,17] and regional hospital that have confirmed on a large regional scale the reduction in TB cases in 2020 vs. 2019 (Department of Epidemiology of Lazio Regional Health Service, unpublished data). Also, from the experience of other epidemics, there are possible explanations [10,11,12,13,14,15,16,17,18,19]. The reorganization of hospital/health services under pressure from COVID-19 resulting in the conversion of tuberculosis wards in COVID-19 unit-had a major indirect impact on the path of care of TB patients, explaining the observed reduction in hospitalization. People with acute and chronic conditions or mild symptoms may have been discouraged from seeking care to reduce crowding in health facilities or fear of getting infected with SARS-CoV-2 in hospital. In addition, the reallocation of healthcare workers and the shift of TB units into COVID-19 units for the pandemic response had an effect in the reduction of health facilities able to diagnosis and treat TB [20,21]. As advised by WHO in TB report 2020 [8], there is a risk that the low notification rate of TB observed, does not means a decrease of incidence but may represents an under-diagnosis likely related to disruption of TB care during first wave of the COVID-19 pandemic. Furthermore, we observed an important and worrying increase of diagnostic delay. In fact, if while in 2019 the time between first symptoms and the start of TB treatment was in average of 38 days, in 2020 this timeframe increased up to a median of 90 days. Noteworthy a previous regional survey indicated 78 days in TB total diagnostic delay [22].

We believe that this increasing is strongly related to the SARS-CoV2 pandemic, the consequent lockdown and health system reorganization due to the high burden of cases of SARS-CoV-2 infection. This is in line with other experiences in the literature documenting an increase in diagnostic delay even for non-communicable diseases [23,24,25]. Comparing the diagnostic delays by each month between 2019 and 2020 (Figure 1), it is possible to highlight that the timeframe between April and July 2020 during lockdown and immediately after was the most critical, while a partial reduction was observed towards August. Diagnostic delays may produce an increase of severity of clinical presentations [26,27]. Indeed, patients diagnosed in 2020 had more severe clinical and radiological findings. They have a higher Timika score, more involved lung lobes and a concomitant diagnosis of extrapulmonary TB. Furthermore, 30% of 2020 patients showed an acute respiratory failure. TB patients in 2020 had lower education level and were more frequently unemployed people vs. 2019. During COVID-19 pandemic there has been a continued contraction of national economies with income reductions. This could have an impact both in the short but especially in the long term on the population that are already fragile and are more susceptible to TB [28].

In multivariate analysis, differences in TB diagnostic delay times between 2019 and 2020 were strengthened with a 7-fold higher risk for diagnostic delay documented in 2020, but the same behavior was not observed for clinical severity at diagnosis. Although the limited number of patients in our study or the length of the observation time impose caution in interpreting these findings, we think that alarm should be raised in rising the awareness of the high cost of diagnostic delay in terms of outcome and patients’ survival. This has been confirmed by several studies [29,30,31] documenting unfavorable outcomes related to advanced stages of the disease at diagnosis, underlined the pivot role that early symptoms recognition, rapid diagnosis and so early treatment have for TB outcome. The year 2020 is also associated with a higher risk of diagnostic delay in a multivariate model that also included education, gender, foreign nationality, occupational status, BMI, presence of comorbidity and TB risk factors. Furthermore, nine out of 115 patients in 2020 also had SARS CoV2 infection. It cannot be excluded that in these patients, SARS-CoV-2 made the clinical status more severe but also hid the tuberculous signs with an increase in diagnostic delay. For this reason, screening for SARS-CoV-2 during the pandemic is recommended in all patients with TB. In addition, we observed an increase in mortality, with one death in 2019 and three deaths in 2020. These data on outcome, although provisional, are a wake-up call for us to consider.

We recognize some limitations in our study: the enrolment of patients diagnosed in one Institution may limit the extent to which our findings can be generalized. The retrospective nature of the study, the lack of data on LTBI, the need to extend the period of evaluation and the continuous evolution of the COVID-19 pandemic with first wave and second wave can precluded the consideration of other factors potentially influencing diagnostic delay and reduction in hospitalizations.

## 4. Materials and Methods

### 4.1. Study Design and Participants

A retrospective data collection and evaluation was conducted to include all the patients hospitalized for TB at INMI from 9 March to 31 August 2020 (lockdown period and three months thereafter). For the purpose of the study data from patients hospitalized in the same period of 2019 were also collected. Data have been extracted from the local TB database approved by L. Spallanzani Institute Ethics Committee (Decision No. 12/2015). All patients provided written informed consent to the utilization of anonymized clinical data. The inclusion criteria were as follows: diagnosis of tuberculosis according to WHO guidelines, age above 18 years [32]. Data collection included patient’s socio-demographic characteristics (i.e., sex, marital status, age, occupational status, education, nationality), risk factors for TB (diabetes, chronic renal failure, malignances, being under immunosuppressive therapy, diabetes, renal failure, HIV infection, previous TB and smoking habit), clinical and microbiological characteristics (symptoms, BMI, drug resistance pattern, concurrent extrapulmonary TB, functional respiratory impairment).

As comorbidities, COPD/bronchiectasis, cardiopathy, hypertension, hypothyroidism, dementia, hematological disease, and chronic liver disease were included. Moreover, data on severity were collected at the admission, by the Timika radiographic score [33,34].

We also evaluated total diagnostic delay (TD) as the time interval between the onset of symptoms and start of TB treatment. This included patient delay (PD) and health system delay (HSD); PD: period from the onset of the first symptom(s) related to pulmonary TB to the first medical consultation; HSD: the time interval between the first medical consultation and start of TB treatment [35,36].

### 4.2. Statistical Analysis

Data were summarized using counts and percentage for categorical variables; and with medians and interquartile ranges were utilized for continuous variables. The data were also stratified by calendar period (2019 vs. 2020) and differences between the two periods were assessed using Mann–Whitney test for continuous measures and the Fisher’s exact test for categorical data.

A preliminary analysis on the HSD comparing the two periods found that there was, on average, an increase of 1 day. Although this increase was statistically significant, we did not consider this delay as clinically significant; we, thus, evaluated only characteristics potentially associated with a long PD, (>30 days) [36], using logistic regression models both univariate and multivariate, with backward selection (*p* = 0.20). The data were analyzed using Stata software, release 15.0. (Stata Corp, College Station, TX, USA).

## 5. Conclusions

In conclusion, our experience suggests that the COVID-19 pandemic has had an impact on TB patient care in terms of higher diagnostic delay, reduced hospitalization, and increased severity of clinical presentations. It is possible that this is the consequence of COVID-19 pandemic and the disruption of TB care services. It should be emphasized that diagnostic delays may produce an increase of *M. tuberculosis’* transmission. The greater effects of COVID-19 pandemic are not yet all visible; therefore, rigorous monitoring involving all national TB center should be perform in order to quantify and prevent them. What is more important to remember is the world will possibly have other pandemics. The priority for all public health is to improve health system resilience to cope with shock events and to be able maintain during crisis the most critical services, including TB one in order to avoid loss of advances in TB control over the past 10 years. Furthermore, a future use of telehealth services can have a pivotal role to guarantee the continuity of care and TB services during a future pandemic.

## Figures and Tables

**Figure 1 antibiotics-10-00272-f001:**
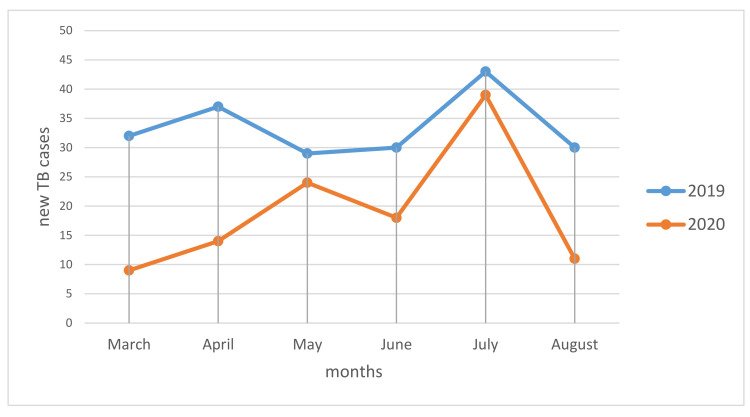
TB cases March August 2019–2020.

**Figure 2 antibiotics-10-00272-f002:**
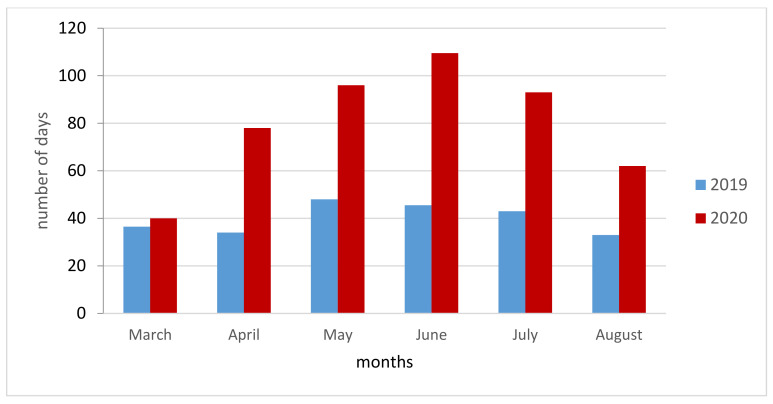
Monthly comparison of Total delay 2019 vs. 2020.

**Table 1 antibiotics-10-00272-t001:** Population characteristics.

		2020 (tot 115)	2019 (tot 201)	*p*-Value
		*n* (%)	*n* (%)	
Sex	Female	39 (34)	59 (29)	0.449
	Male	76 (66)	142 (71)	
BMI	Low (16–18,49)	66 (57)	68 (34)	<0.01
	Normal (18.5–24,99)	39 (34)	121 (60)	
	High (25–29,99)	10 (9)	12 (5.5)	
	Very high (>29.99)	5(4)	1(0.5)	
Marital Status	Single	50 (43)	72 (36)	<0.01
	Married	60 (52)	91 (45)	
	Not declared	5 (4)	38 (19)	
Age	18–40	41 (36)	78 (39)	0.709
	41–64	60 (52)	95 (47)	
	>65	14 (12)	28 (14)	
Occupational status	Employed	33 (29)	89 (44)	0.013
	Unemployed	72 (63)	92 (46)	
	Retired	10 (9)	20 (10)	
Education	<8 years	106 (92)	168 (84)	0.038
	>8 years	9 (8)	33 (16)	
Nationality	African	15 (13)	29 (14)	0.99
	Central-Sud American	9 (8)	18 (9)	
	Asian	18 (16)	32 (16)	
	East European	35 (30)	60 (30)	
	Italian	38 (33)	62 (31)	
Smoking	Yes	64 (56)	70 (35)	<0.01

**Table 2 antibiotics-10-00272-t002:** Clinical characteristics.

		2020 (tot 115)	2019 (tot 201)	*p*-Value
		*n* (%)	*n* (%)	
Comorbidity	Yes	63 (56)	94 (47)	0.160
Previous TB	Yes	16 (14)	24 (12)	0.603
Risk factor for TB	Diabetes	14 (12)	20 (10)	0.574
	Hypertension	16 (14)	27 (13)	1.000
	Renal failure	5 (4)	13 (6)	0.615
	HIV positive	5 (4)	3 (1)	0.145
Initial TB symptoms	Cough	81 (70)	130 (65)	0.322
	Fever	27 (23)	66 (33)	0.095
	Dispnea	26 (23)	17 (8)	<0.01
	Night sweats	27 (23)	31 (15)	0.096
	Hemoptysis	11 (10)	30 (15)	0.223
	Weight loss	53 (46)	56 (28)	<0.01
TB cases	Bacteriologically confirmed	94 (82)	163 (81)	1.000
	Clinically diagnosed	21 (18)	38 (19)	
Timika score	Timika 1 (≤60)	45 (39)	134 (67)	<0.01
	Timika 2 (>60; ≤100)	51 (44)	57 (28)	
	Timika 3 (>100)	19 (17)	10 (5)	
Sputum smear	Positive	55 (48)	82 (41)	0.240
Acute respiratory failure	Yes	34 (30)	17 (8)	<0.01
Drug resistance	Susceptible	94 (82)	176 (88)	0.185
	Monoresistance	10 (9)	12 (6)	0.036
	Polydrug resistance	1 (1)	2 (1)	1.000
	Multidrug resistance	9 (7)	6 (3)	0.059
Concurrent extrapulmonary TB	Yes	37 (32)	31 (15)	<0.01

**Table 3 antibiotics-10-00272-t003:** Diagnostic delay.

	2020 (tot 115)	2019 (tot 201)	*p*-Value
	*n* (%)	*n* (%)	
Patient delay, days (median; IQR)	75 (40–100)	30 (10–60)	<0.01
HS, days (median; IQR)	5 (3–14)	4 (2–10)	0.032
Total, days (median; IQR)	90 (58–107)	38 (22–69)	<0.01

**Table 4 antibiotics-10-00272-t004:** Patients with a patient delay (PD) > 30 days, stratified by several patient characteristics.

		PD > 30 Days			
		N	%	Total	*p*-Value
Sex	Male	118	54.1	218	0.807
	Female	55	56.1	98	
Marital status	Single	72	59	122	0.084
	Married	84	55.6	151	
	Not indicated	17	39.5	43	
Age class	18–40	66	55.5	118	0.987
	41–64	84	54.2	155	
	>65	23	54.8	42	
Education	<8 years	159	58	274	0.004
	>8 years	14	33.3	42	
Nationality	Italian	43	43	100	0.005
	Foreign	130	60.2	216	
Occupational status	Employed	66	54.1	122	0.957
	Unemployed	91	55.5	164	
	Retired	16	55.3	30	
Smoke habit	Yes	72	53.7	134	0.819
	No	101	55.5	182	
Year	2020	93	80.9	115	<0.01
	2019	80	39.8	201	
BMI	<18.5	88	65.7	134	<0.01
	≥18.5	85	46.7	182	
Comorbidity	Yes	94	59.5	158	0.113
	No	79	50	158	
Diabetes	Yes	20	58.8	34	0.716
	No	153	54.3	282	
Hypertension	Yes	23	53.5	43	0.871
	No	150	55	273	
Renal failure	Yes	6	33.3	18	0.086
	No	167	56	298	
HIV positive	Yes	6	75	8	0.301
	No	167	54.2	308	
Acute respiratory failure	Yes	38	74.5	51	0.002
	No	155	50.9	265	
Previous TB	Yes	22	55	40	1.000
	No	151	54.7	276	

**Table 5 antibiotics-10-00272-t005:** Association of some characteristics with a patient delay (PD) > 30 days.

		OR	*p*-Value	95%CI		AOR	*p*-Value	95%CI	
Sex	Male	0.92	0.742	(0.57–1.49)		NI			
	Female	1.00							
Marital status	Single	1.14	0.574	(0.71–1.86)		NI			
	Married	1.00							
	Not indicated	0.52	0.065	(0.26–1.04)					
Age class	18–40	1.00				NI			
	41–64	0.95	0.834	(0.59–1.54)					
	>65	0.97	0.937	(0.48–1.97)					
Education	<8 years	1.00				NI			
	>8 years	0.36	0.004	(0.18–0.72)					
Nationality	Italian	1.00				1.00			
	Foreign	2.00	0.005	(1.24–3.24)		2.93	0.000	1.65	5.21
Occupational status	Employed	1.00				NI			
	Unemployed	1.06	0.815	(0.66–1.69)					
	Retired	0.97	0.940	(0.44–2.16)					
Smoke habit	Yes	0.93	0.756	(0.59–1.46)		NI			
	No	1.00							
Year	2020	6.39	0.000	(3.71–11.01)		6.93	0.000	3.91	12.30
	2019	1.00							
BMI	<18.5	2.18	0.001	(1.38–3.46)		NI			
	≥18.5	1.00							
Comorbidity	Yes	1.47	0.091	(0.94–2.29)		2.05	0.010	1.19	3.53
	No	1.00							
Diabetes	Yes	1.20	0.614	(0.59–2.48)		NI			
	No	1.00							
Hypertension	Yes	0.94	0.858	(0.49–1.80)		NI			
	No	1.00							
Renal failure	Yes	0.39	0.068	(0.14–1.07)		0.32	0.048	0.11	0.99
	No	1.00							
HIV positive	Yes	2.53	0.260	(0.50–12.75)		NI			
	No	1.00							
Acute respiratory failure	Yes	2.81	0.003	(1.43–5.52)		NI			
	No	1.00							
Previous TB	Yes	1.01	0.973	(0.52–1.97)		NI			
	No	1.00							

NI: not included.

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
