# Peer review of "Increase in Tuberculosis Diagnostic Delay during First Wave of the COVID-19 Pandemic: Data from an Italian Infectious Disease Referral Hospital"

_antibiotics, 2021, doi:10.3390/antibiotics10030272_

Round 1

Reviewer 1 Report

Increase in Tuberculosis delay during first wave of COVID-19 pandemic: data from an Italian Infectious Disease Referral Hospital

Dear author and editor:

This study talked about the problem of TB diagnostic delay during the first wave of COVID19  pandemic by retrospective data collection at the national institute for infectious disease in Rome (L.spallanzani). The authors focused on important problem that the low rates of TB cases during this period does not mean decreased incidence but it because of delay in diagnosis and may increase the risk of TB transmission. 

This article could be published after a minor revision and i have some comments on it: 

  • Is there any data about co-infected patients (TB-COVID19)? could the co-infection be one of the reasons of delay in TB diagnosis?
  • Do you think that the quarantine and the lockdown procedures had a negative impact on TB diagnosis?
  • In the figure 1 and 2 you should insert (days) on Y axis and month on X axis, the figure legends have to be written under each figure not above.
  • Is there any similar study in other Italian hospital or infectious disease centre from the same region?
  • Is there a difference between the percentage of symptomatic active TB cases to LTB cases during first wave of COVID19 pandemic respect to pre-COVID19 period?

Thank you very much, Best regards

Author Response

To Antibiotics Editor and reviewers,

We have appreciated the positive feedback to our manuscript “Increase in Tuberculosis diagnostic delay during first wave of COVID-19 pandemic: data from an Italian Infectious Disease Referral Hospital”.

We have considered all the useful suggestions made by the referees and we have implemented the text. We have also satisfied the technical requirements according to the journal guidelines. Modifications have been highlighted using "track changes" feature. Also, a native English speaker has been engaged to improve the fluency and the readability of the manuscript.

We believe that the revision proposed by the referees, and further implemented in the text, contributed to improve the manuscript. Thus, we kindly ask to and re-consider the manuscript for publication.

Please find a point-by-point response to the referees’ comments below.

Best regards,

Dr. Francesco Di Gennaro

Reviewer's Responses to Questions

Reviewer 1

This study talked about the problem of TB diagnostic delay during the first wave of COVID19  pandemic by retrospective data collection at the national institute for infectious disease in Rome (L.spallanzani). The authors focused on important problem that the low rates of TB cases during this period does not mean decreased incidence but it because of delay in diagnosis and may increase the risk of TB transmission. 

Response

We thank you very much for the encouraging feedback on our manuscript.  Below the answer for your interesting and appropriate questions.

Questions

This article could be published after a minor revision and i have some comments on it: 

  • Is there any data about co-infected patients (TB-COVID19)? could the co-infection be one of the reasons of delay in TB diagnosis?

Response

  • Yes, we have data on TB-COVID coinfection. Eight of 115 patients had TB-Covid 19 co-infections.  SARS CoV2 may have influenced the diagnostic delay of these patients hiding signs and symptoms (published data Active tuberculosis, sequelae and COVID-19 co-infection: first cohort of 49 cases. Eur Respir J. 2020 Jul 9;56(1):2001398. and Concurrent cavitary pulmonary tuberculosis and COVID-19 pneumonia with in vitro immune cell anergy. Infection. 2021 Jan 17:1–4). Therefore, coherently with your suggestion, we have added the following sentence to the text: "Nine out of 115 patients in 2020 also had SARS CoV2 infection. It cannot be excluded that in these patients SARS-CoV-2 has made the clinical status more severe but also hidden the tuberculous signs with an increase in diagnostic delay. For this reason, screening for SARS-CoV-2 during the pandemic is recommended in all patients with TB ".

Question

- Do you think that the quarantine and the lockdown procedures had a negative impact on TB diagnosis?

Response

Thank you a lot. We think that no quarantine or lockdown had a direct negative impact on TB diagnoses but rather the reorganization and  conversion of most hospital wards in COVID-19 unit. As stated in the paper, due to high COVID 19 burden, Italian National Health System was forced to reorganize itself with conversion of most hospital wards in COVID-19 unit, health workforce was reallocated and most of the non-urgent services were suspended. In this scenario, most of wards and outpatients’ services for TB of Italian hospital were also interrupted in their routine activities, as happened in our hospital that is an Italian TB referral hospital.

In fact we concluded as follow: Priority for all public health is to improve health system resilience to cope with shock events and to be able to maintain during crisis the most critical services, including TB one in order to avoid loss of advances in TB control over the past 10 years.

Question

-In the figure 1 and 2 you should insert (days) on Y axis and month on X axis, the figure legends have to be written under each figure not above.

Response

Thank you a lot for these suggestions. We modified following your indications

Question

-Is there any similar study in other Italian hospital or infectious disease centre from the same region?

Response

Unfortunaly no, but a preliminary regional data shown a reduction of hospitalitation for TB in all   Lazio region's hospital

Question

- Is there a difference between the percentage of symptomatic active TB cases to LTB cases during first wave of COVID19 pandemic respect to pre-COVID19 period?

Response

Thank you for your sharable question. We did not evaluate pre- and post-pandemic covid cases of LTBI. We add this within the limitations of study and can be a future perspectives for our study

We add in our discussion the following sentences: “ We recognize some limitations in our study: the enrolment of patients diagnosed in one Institution may limit the extent to which our findings can be generalized. The retrospective nature of the study, the lack of data on LTBI,  the need to extend the period of evaluation and the continuous evolution of the COVID-19 pandemic with first wave and second wave can precluded the consideration of other factors potentially influencing diagnostic delay and reduction in hospitalizations.

Reviewer 2 Report

The authors present a paper titled: "Increase in Tuberculosis diagnostic delay during first wave of COVID-19 pandemic: data from an Italian Infectious Disease Referral Hospital" that I found very interesting from a clinical point of view for the present pandemic situation and for remembering how some diseases are relevant despite the pandemia. Each disease should maintain a respectful dignity.  Introduction is clear as well as the collected patients as number and homogeneity. 

I have some problems in understanding the sequence of the chapters and the rational for keeping this sequence:

1.Introduction  2. Results  3. Discussion. 4. Materials and Methods. 5. Conclusions

Furthermore I ask the Authors to give me some more precise information about the outcome of the 2 population. 2020 (115), 2019 (201). They report that a significant increase in diagnostic delay was observed : 75 days in 2020 compared to 30 days in 2019. I understood that this delay created a serious clinical deterioration. What about mortality or disabling condition? I am interested in the outcome of these patients.

table 4 and 5 should be added as appendix, not in the text.

Conclusions: the Authors should make an effort to give some practical suggestions to overcome the problem and this could be a good example also for other diseases during this pandemia

Author Response

To Antibiotics Editor and reviewers,

We have appreciated the positive feedback to our manuscript “Increase in Tuberculosis diagnostic delay during first wave of COVID-19 pandemic: data from an Italian Infectious Disease Referral Hospital”.

We have considered all the useful suggestions made by the referees and we have implemented the text. We have also satisfied the technical requirements according to the journal guidelines. Modifications have been highlighted using "track changes" feature. Also, a native English speaker has been engaged to improve the fluency and the readability of the manuscript.

We believe that the revision proposed by the referees, and further implemented in the text, contributed to improve the manuscript. Thus, we kindly ask to and re-consider the manuscript for publication.

Please find a point-by-point response to the referees’ comments below.

Best regards,

Dr. Francesco Di Gennaro

Reviewer's Responses to Question

Reviewer 2

The authors present a paper titled: "Increase in Tuberculosis diagnostic delay during first wave of COVID-19 pandemic: data from an Italian Infectious Disease Referral Hospital" that I found very interesting from a clinical point of view for the present pandemic situation and for remembering how some diseases are relevant despite the pandemia. Each disease should maintain a respectful dignity.  Introduction is clear as well as the collected patients as number and homogeneity. 

Response:

We thank you very much for your  feedback on our manuscript.  We have considered all the useful suggestions to improve our research. Below point by point response. Furthermore, A native English speaker has been engaged to improve the fluency and the readability of the manuscript.

Question

I have some problems in understanding the sequence of the chapters and the rational for keeping this sequence:

1.Introduction  2. Results  3. Discussion. 4. Materials and Methods. 5. Conclusions

Response

We are sorry for that but we followed the journal’s formatting guidelines and we cannot change

- Furthermore I ask the Authors to give me some more precise information about the outcome of the 2 population. 2020 (115), 2019 (201). They report that a significant increase in diagnostic delay was observed : 75 days in 2020 compared to 30 days in 2019. I understood that this delay created a serious clinical deterioration. What about mortality or disabling condition? I am interested in the outcome of these patients.

Response:

We greatly appreciate your observation. The increase in diagnostic delay has caused and is causing even in patients that we are seeing more serious clinical conditions with a greater number of lung lobes involved, a higher Timika score, more cases of respiratory failure etc. In our period of study we observed 1 death in 2019 and 3 in 2020. We added in our text this data coherently with your suggestion: “In add, we observed an increase in mortality, with one death in 2019 and three deaths in 2020. These data on outcome, although provisional, are a wake-up call to consider.”

Question

table 4 and 5 should be added as appendix, not in the text

Response

Thanks a lot for the observation. Also here we have followed the formatting guidelines of the journal so we cannot change

Question

-Conclusions: the Authors should make an effort to give some practical suggestions to overcome the problem and this could be a good example also for other diseases during this pandemia

Response:

Thank you a lot for your suggestions. We clarify in our article how the impact of COVID-19 pandemic on TB services was estimated to be dramatic, especially in countries where healthcare staff involved in TB management have been reassigned to the COVID-19 emergency. WHO advised that the disruption of TB services, due to the COVID-19 pandemic, could lead to fewer TB diagnoses, an increase in diagnostic delay and TB death.

Coherent with this and following your suggestion we add in our conclusion:

“Priority for all public health is to improve health system resilience to cope with shock events and to be able maintain during crisis the most critical services, including TB one in order to avoid loss of advances in TB control over the past 10 years.

Furthermore a future use of telehealth services can has a pivot role to guarantee continue of care and TB services during future pandemic”

Reviewer 3 Report

Thank you for an interesting manuscript, investigating the important challenge of delayed diagnosis and treatment for other diseases (in this case TB) during the COVID-pandemic. I have some major and minor points:

Major:

General: While the manuscript is well-structured and presented, it contains a number of gramatical errors (e.g. missing "the"/"a") and typos (e.g. missing spaces), so I would suggest an extra pass of language checking.

Abstract line 34: It is in the abstract unclear, what the comparision grouo (patients from 2019) were. This is very clear in the manuscript itself, ut should also be made clear in the abstract.

Abstract & keywords 40-47:  You forgot to remove some of the template text, making this part very confusing.

Table 4: This table would be more clear if 

Line 202-205: This statement seems improbably, why povertry generally increased during the pandemic, it is unrealistic that low eductation should have increased, as persons education in general doesnøt go down on an individual level (and with only 1 year time horizon a change in population education cannot be the explanation). Hence, you should reconsider this statement. Probably the lower educated patients are now a larger group of the TB patients?

Line 210: It sounds as if you carried out a regression analysi for clinical severity, but you donøt report the results apart from this statement in the discussion.

Discussion: To which degree could the later diagnosis of TB be explained by patients with TB being referred and tested for COVID-19 before it (later) is observed, that the cause for the symptoms is TB? Maybe you can see to which this phenomenon could be part of the additional delay (there is anecdeotcal evidence that this has been a problem in certain areas in Northern Europe).

Line 217-219: This is a repetition from the start of the paragraph-

Line 226-236: This section is repeated in the discussion

Minor:

Tables: What does the * after "p-value" indicate?

Line 107: A very brief explanation of the definition of "diagnostic delay" here would be helpfull, it is explained in the methods section, but due to the order fo sections, a short statement here would be good.

Line 115: What is "PD"?

Table 5: I suspect that "NI" stands for "Not included", but this is not clearly specified. Furthermore, the table would be easier to read if the CI were as one column, e.g. (0.57; 1.49)

Line 258: It seems like you use HS and HSD as abbreviations for the same concepts.

Line 285-286: It seems like the supplemnatl figure is paret of the main document.

Line 287-294: The author contributions are not specified.

Author Response

To Antibiotics Editor and reviewers,

We have appreciated the positive feedback to our manuscript “Increase in Tuberculosis diagnostic delay during first wave of COVID-19 pandemic: data from an Italian Infectious Disease Referral Hospital”.

We have considered all the useful suggestions made by the referees and we have implemented the text. We have also satisfied the technical requirements according to the journal guidelines. Modifications have been highlighted using "track changes" feature. Also, a native English speaker has been engaged to improve the fluency and the readability of the manuscript.

We believe that the revision proposed by the referees, and further implemented in the text, contributed to improve the manuscript. Thus, we kindly ask to and re-consider the manuscript for publication.

Please find a point-by-point response to the referees’ comments below.

Best regards,

Dr. Francesco Di Gennaro

Reviewer's Responses to Question

Reviewer 3

 Thank you for an interesting manuscript, investigating the important challenge of delayed diagnosis and treatment for other diseases (in this case TB) during the COVID-pandemic. I have some major and minor points:

-Major:

General: While the manuscript is well-structured and presented, it contains a number of gramatical errors (e.g. missing "the"/"a") and typos (e.g. missing spaces), so I would suggest an extra pass of language checking.

Response:

We thank you very much for the encouraging feedback on our manuscript.  We have considered all the useful suggestions to improve our research. A native English speaker revised the manuscript.

Below point by point response.

Question

- Abstract line 34: It is in the abstract unclear, what the comparision grouo (patients from 2019) were. This is very clear in the manuscript itself, ut should also be made clear in the abstract.

Abstract & keywords 40-47:  You forgot to remove some of the template text, making this part very confusing.

Response:

Thank you a lot for your suggestions. We modified the abstract and keywords and delete the template text

Question

- Line 202-205: This statement seems improbably, why povertry generally increased during the pandemic, it is unrealistic that low eductation should have increased, as persons education in general doesnøt go down on an individual level (and with only 1 year time horizon a change in population education cannot be the explanation). Hence, you should reconsider this statement. Probably the lower educated patients are now a larger group of the TB patients?

Response:

Thank you for your shareable observation. We think that the economic crisis that has occurred due to the covid pandemic can indirectly affect both in the short but especially in the long term for those parts of the population that,already fragile and poor, are more susceptible to tuberculosis. Therefore we have clarified our concept better and following your suggestion added to the text: "

TB patients in 2020 had lower education level and were more frequently unemployed people vs 2019. During COVID-19 pandemic there has been a continued contraction of national economies with income reductions. This could have an impact both in the short but especially in the long term on the population that are already fragile and are more susceptible to TB [28].

Question

- Line 210: It sounds as if you carried out a regression analysi for clinical severity, but you donøt report the results apart from this statement in the discussion.

Response:

Thank you for your suggestions. In table 2 and results section we clarify better: Clinical characteristics are shown in Table 2. During 2020 vs 2019 period patients with dyspnea (23% vs. 8%), weight loss (46% vs. 28%), with acute respiratory failure (30% vs. 8%) and concurrent extra pulmonary TB (32% vs. 15%), were significantly more frequently hospitalized.

Timika score showed a significant worse radiographic pattern in 2020 patients vs 2019 group. (p=<0.01).

Question

Discussion: To which degree could the later diagnosis of TB be explained by patients with TB being referred and tested for COVID-19 before it (later) is observed, that the cause for the symptoms is TB? Maybe you can see to which this phenomenon could be part of the additional delay (there is anecdeotcal evidence that this has been a problem in certain areas in Northern Europe).

Response:

Nine out of 115  patients had TB-Covid 19 co-infections.  SARS-CoV-2 may have influenced the diagnostic delay of these patients hiding signs and symptoms (published data Active tuberculosis, sequelae and COVID-19 co-infection: first cohort of 49 cases. Eur Respir J. 2020 Jul 9;56(1):2001398. and Concurrent cavitary pulmonary tuberculosis and COVID-19 pneumonia with in vitro immune cell anergy. Infection. 2021 Jan 17:1–4). Therefore, coherently with your suggestion, we added the following sentence to the text: " Nine out of 115 patients in 2020 also had SARS CoV2 infection. It cannot be excluded that in these patients SARS-CoV-2 has made the clinical status more severe but also hidden the tuberculous signs with an increase in diagnostic delay. For this reason, screening for SARS CoV2 during the pandemic is recommended in all patients with TB ".

Question

Minor:

Tables: What does the * after "p-value" indicate?

Response:

Thank you. We delete *

Question

-Line 107: A very brief explanation of the definition of "diagnostic delay" here would be helpfull, it is explained in the methods section, but due to the order fo sections, a short statement here would be good.

Response:

Thank you for this suggestion. We modify as following :

A significant increase in total diagnostic delay (TD, time interval between the onset of symptoms and start of TB treatment ) was observed. The median patient delay (PD, period from the onset of the first symptom(s) related to pulmonary TB to the first medical consultation)  was 75 days (IQR: 40-100) in 2020 compared to 30 days (IQR: 10-60) in 2019 (p<0.01). Delay by month of admission in  2020 was higher for all the months considered, except for March, compared with the same months of 2019 (Figure 2). The healthcare system (HSD, the time interval between the first medical consultation and start of TB treatment) delay was 5 days (IQR: 3-14) in 2020 versus 4 (IQR: 2-10) in 2019; the total delay (PD+HSD) was 90 days (IQR: 58-107) and 38 (IQR: 22-69), in 2020 and in 2019, respectively (p<0.01) (Table 3).

Question

Line 115: What is "PD"?

Response:

Thank you we clarify PD (Patients delay), HSD (Health system delay), TD (total delay, PD+HSD) and We standardized these definitions in the text

Question:

Table 5: I suspect that "NI" stands for "Not included", but this is not clearly specified. Furthermore, the table would be easier to read if the CI were as one column, e.g. (0.57; 1.49)

Response:

We add a short legend at the end of the table where NI: Not included

Question

Line 258: It seems like you use HS and HSD as abbreviations for the same concepts.

Response:

 Thank you, we uniformy this in the text

Response:

Line 285-286: It seems like the supplemnatl figure is paret of the main document.

Response:

Thank you for your suggestion. We delete it and we left the figure in the text and. no more in the supplementary material      

Question

Line 287-294: The author contributions are not specified.

Response:

 Thank you for your suggestion. We wrote the author contributions as following:” “Conceptualization, F.P., G.G., F.D.G; methodology, E.G, L.T, G.I.; formal analysis, L.T, E.G, F.D.G; data curation, S.M., P.V., R.L., C.C., L.P., C.N., S.I., N.B., F.I., A.M., S.T., D.G.; writing—original draft preparation, F.D.G, G.G, F.P, E.G., L.T.; writing—review and editing, G.I., E.G., F.P.; supervision, F.V., G.I. All authors have read and agreed to the final version of the manuscript.”